# Empirically calibrated simulations reveal the limits of phenotypic clustering algorithms for biodiversity assessment in data-scarce crops

Abdel Kader Naino Jika ◉*

Faculty of Agronomy, Department of crops productions, University Abdou Moumouni, Niamey, Niger

* kaderjika@gmail.com

## Abstract

Clustering algorithms are widely used for phenotypic characterization and germplasm management, particularly in data-scarce crops such as neglected and underutilized species (NUS) that lack genomic resources. However, their performance under biologically realistic conditions remains poorly understood. Standard clustering methods commonly applied in crop research often assume distinct, isotropic, and homogeneous clusters, assumptions rarely satisfied in real-world phenotypic data-sets. We developed a flexible and empirically calibrated simulation framework, using phenotypic data from West African fonio (*Digitaria exilis*), to benchmark the performance of eleven clustering algorithms under both idealized and realistic scenarios. Our simulations integrated heterogeneous trait distributions (normal, gamma), strong inter-trait correlations (up to $r=-0.84$), heteroscedasticity, and moderate population structure (mean $Pst=0.16\pm0.001$, achieved through iterative calibration). Each scenario was replicated 100 times, with clustering accuracy evaluated using external (ARI, NMI) and internal (Silhouette, Davies–Bouldin) validation metrics under standardized conditions. The results revealed consistently poor algorithm performance under realistic conditions (e.g., ARI $< 0.07$), including for widely used methods in Neglected and Underutilized Species (NUS) research such as K-means, GMM, and PAM. Notably, conventional validation metrics failed to detect biologically meaningful structure revealed by geometric diagnostics, highlighting a critical methodological limitation. Performance markedly improved under idealized conditions, validating our simulation framework. These findings highlight the risk of overinterpreting clustering outputs from weakly structured phenotypic datasets and expose key limitations in current biodiversity analysis practices, particularly those guiding plant genetic resource conservation programs. We provide an open-source R-based diagnostic tool, with parameter specifications to assist practitioners in selecting reproducible and interpretable clustering approaches for germplasm management and biodiversity assessment in data-scarce crops.

**Data availability statement:** All empirical parameters used for simulation calibration were extracted from [1]. Agromorphological characterization revealed three phenotypic groups in a region-wide germplasm of fonio (Digitaria exilis (Kippist) Stapf) from West Africa. Agronomy, 10(11), 1653., whose full dataset is publicly available within the article and its supplementary tables. All relevant data are within the manuscript and its Supporting information files. In addition, the complete R script used to simulate phenotypic datasets, apply clustering algorithms, and compute evaluation metrics is publicly available on Zenodo: https://doi.org/10.5281/zenodo.15877863. This ensures full reproducibility of the results presented.

**Funding:** The author(s) received no specific funding for this work.

**Competing interests:** The author has declared that no competing interests exist.

## Introduction

Neglected and underutilized species (NUS) constitute a critical reservoir of agricultural biodiversity, essential for global food security, sustainable agriculture, and adaptation to climate change and ecological instability [2,3]. Despite their resilience and adaptive potential, many NUS—such as fonio (*Digitaria exilis*), a cereal cultivated in ecologically marginal environments—remain underrepresented in genomic databases due to limited funding and molecular resources [1,4]. Consequently, scientists and breeders often rely on phenotypic data as a primary resource for exploring genetic structure, managing germplasm collections, and informing breeding strategies [5].

Yet, phenotypic analyses face fundamental challenges. These include subtle population differentiation and complex data distributions—characteristics often overlooked in methodological practice [6,7]. Fonio epitomizes this issue: historically vital to marginalized West African communities, it thrives in agroecological niches unsuitable for major cereals [1,8]. In the absence of genomic profiling, phenotypic clustering is often the only available framework guiding biodiversity assessments and conservation interventions [9].

Crucially, the interpretability of phenotypic clustering depends on assumptions that are rarely met in NUS datasets—namely, that clusters are discrete, isotropic, and internally homogeneous [10,11]. In practice, phenotypic traits in NUS often exhibit overlapping distributions, strong inter-trait correlations, heteroscedasticity, and low genetic differentiation (Pst ≈ 0.1–0.15), reflecting continuous gene flow and mixed mating systems [12–14].

Despite these biological complexities, most benchmarking studies assume idealized conditions. This disconnect raises concerns about the validity of widely used clustering metrics—such as Adjusted Rand Index (ARI) and Normalized Mutual Information (NMI)—which are sensitive to violations of assumptions like balanced sample sizes, distinct separation, and homogeneous variance [12,15]. While methodological critiques have emerged [13], few empirical studies have systematically quantified how these biases affect clustering outcomes in realistic biological scenarios. This gap is structural, not merely technical.

Several popular algorithms illustrate this misalignment:

- **K-means** and **PAM** assume spherical clusters with equal variances—conditions rarely met in empirical datasets [11];

- **Ward's method** favors compact, globular clusters but collapses under trait-driven asymmetries [16];

- **Gaussian Mixture Models (GMM)** offer flexibility but still assume normality, often violated by real traits [17];

- **DBSCAN** accommodates arbitrarily shaped clusters but fails when densities vary across groups [18,19].

These algorithmic limitations have real-world consequences. Misclassification can result in the exclusion of locally adapted landraces or the loss of drought-resilient accessions—missed opportunities that are particularly costly in resource-limited

settings. This structural misalignment directly undermines biodiversity conservation targets. As demonstrated by Arneth et al. [20], climate pressures threaten to invalidate preservation strategies for NUS based on artificial phenotypic clusters, even when controlling for non-climatic pressures like habitat exploitation and ecological fragmentation.

To address this methodological gap, we introduce a biologically grounded simulation framework tailored to the statistical and ecological realities of neglected and underutilized species (NUS). Calibrated with empirical phenotypic data from West African fonio (Digitaria exilis), the framework enables rigorous benchmarking of eleven clustering algorithms—including K-means, PAM, Ward's method, GMM, DBSCAN, HDBSCAN, Spectral clustering, Fuzzy C-means, Affinity Propagation, Self-Organizing Maps (SOM), and Two-Step clustering—under both idealized and realistic conditions. The simulated datasets reproduce key real-world complexities: heterogeneous trait distributions (normal and gamma), strong inter-trait correlations (up to $r = -0.84$), heteroscedastic variances, and moderate population structure ($Pst \approx 0.15$). We evaluate algorithm performance across 100 simulation replicates using ARI, NMI, Silhouette coefficient, and Davies–Bouldin index, supplemented by dimensionality reduction tools such as PCA and UMAP to assess the geometric structure of clusters.

Our central hypothesis is that apparent clusters in phenotypic datasets of NUS with modest genetic structure often arise as statistical artefacts driven by methodological constraints, rather than reflecting genuine biological partitions. By testing this systematically, our study offers a reproducible, R-based diagnostic tool to evaluate clustering validity before drawing evolutionary or agronomic conclusions.

We advocate for a paradigm shift: empirically calibrated simulations should replace idealized Gaussian models as the standard for evaluating clustering performance in NUS research. This shift prioritizes biological realism over computational convenience, particularly in regions such as sub-Saharan Africa, where genotyping is often inaccessible, while phenotyping remains more feasible and cost-effective. The tool is open-source, modular, and designed for reproducibility and cross-species adaptability. To maximize generalizability, its parameters were not only calibrated from fonio, but also designed to accommodate empirical data from other NUS available in the literature.

This article thus serves as the methodological cornerstone of a broader research agenda aimed at systematically benchmarking clustering approaches for biodiversity assessment across diverse underutilized taxa. It promotes a phenotype-first paradigm in agrobiodiversity research—one that centers ecological realism, supports data-driven decision-making, and advances conservation and breeding efforts in resource-limited settings. Although calibrated using fonio as a model, the framework is crop-agnostic and can be readily reparameterized for other neglected species by adjusting the trait covariance structure, differentiation level (Pst), and distributional assumptions. This flexibility makes it suitable as a general diagnostic platform for phenotypic clustering in data-scarce crops.

## Materials and methods

### Empirical data and trait architecture

This study draws upon agromorphological trait data from 180 *Digitaria exilis* (fonio) landraces characterized by Bio et al. [1], spanning three geographically distinct groups (n = 91, 43, and 46). Eight traits were selected for their agronomic relevance and quantitative tractability: plant height (PHT), number of internodes (NIN), flowering time (FLO), maturity time (MAT), number of grains per inflorescence (NGR), grain yield per plant (GRY), harvest index (HI), and thousand-seed weight (TSW). All empirical parameters were manually extracted from the published tables of Bio et al. [1], *Journal of Crop Improvement* 34(4):512–530. Specifically, group sizes, trait means, standard deviations, and the inter-trait correlation matrix were taken directly from the reported agromorphological dataset. These empirical statistics were used to reconstruct the phenotypic covariance structure and served as calibration inputs for the simulation of realistic fonio populations.

Trait distributions were predominantly normal, with the exception of GRY, which followed a gamma distribution. Empirical trait correlations (range: $r = -0.84$ to $0.77$) were encoded in a near-positive definite correlation matrix [21], and group-specific variances were retained to reproduce heteroscedastic structures in simulations.

## Simulation framework

**Phenotypic differentiation (P*st*) estimation and calibration.** Phenotypic differentiation ($P_{st}$) was computed as the proportion of total phenotypic variance explained by between-group differences, analogous to Wright's $F_{st}$ for quantitative traits [22]. The phenotypic covariance matrices used for Pst calibration were derived from the empirical correlation structure of Bio et al. [1], ensuring that simulated trait dependencies reproduce the inter-trait architecture observed in real fonio landraces.

For a trait matrix $X$ with group labels $g$, $P_{st}$ was calculated as:

$$P_{st} = \frac{V_B}{V_T}$$

where $V_B$ is the trace of the between-group covariance matrix (inter-group variance component) and $V_T$ is the trace of the total covariance matrix.

In simulations, the between-group covariance matrix was iteratively adjusted using a scaling factor until the realized $P_{st}$ converged to the target value—0.15 for the realistic scenario and 0.80 for the idealized scenario. Convergence was accepted when $|\,P_{st}^{realized} - P_{st}^{target}\,| < 0.01$. Across 100 replicates, the mean realized $P_{st}$ was $0.16 \pm 0.001$ for the realistic scenario, confirming accurate calibration.

This procedure ensures that the simulated datasets reproduce the moderate phenotypic differentiation observed in fonio landraces [1], while maintaining the framework's flexibility for other neglected crops by allowing adjustment of the target $P_{st}$ parameter. The calibration procedure is implemented in the function simulate_populations(), which computes $P_{st}$ through the helper function calc_fst_pheno().

**Rationale for the P*st* target and comparability with F*st*.** The realistic-scenario target of P*st* ≈ 0.15 was chosen to represent a biologically plausible level of moderate differentiation commonly observed in smallholder and landrace systems, where recurrent seed exchange, open pollination, and overlapping agroecological conditions limit genetic divergence. Conceptually, P*st*—the proportion of phenotypic variance attributable to between-group differences—is an observable proxy for Q*st* when heritabilities are unknown. However, comparisons to neutral-marker F*st* must be interpreted with caution, because P*st* also depends on environmental variance and the ratio of additive genetic variances within versus among groups [23,24].

Our choice of approximately 0.15 thus reflects a heuristic midpoint rather than a strict numerical equivalence to genetic differentiation, but it aligns closely with the low-to-moderate genetic structure recently quantified in fonio. In a genomic study of *Digitaria exilis* populations across Senegal, Diop et al. [25] reported a mean F*st* = 0.054 (range −0.038 to 0.18), indicating weak but significant differentiation among ethnolinguistic groups. This pattern reflects shared evolutionary and management processes—continuous gene flow, recurrent seed exchange, and stabilizing selection under comparable ecological conditions—typical of farmer-managed landrace systems in West Africa.

Importantly, Diop et al. [25] demonstrated that social and cultural boundaries (ethnic and linguistic groups) explained more of the genetic structure than geography or environment, and that the highly self-pollinating mating system of fonio (observed heterozygosity ≈ 0.05) contributes to maintaining intra-group uniformity while preserving subtle inter-group divergence. These findings reinforce the biological realism of our simulation framework, in which moderate phenotypic differentiation arises from socially mediated but genetically connected populations.

We therefore explicitly avoid relying on earlier studies that reported inflated differentiation values based on dominant markers (e.g., AFLP), which are unsuitable for calibrating a low-structure regime. Instead, our empirical grounding in Diop et al. [25] ensures that the simulated scenarios faithfully represent the weakly structured, socially stratified population architecture characteristic of neglected and underutilized crops.

Two simulation scenarios were implemented to assess clustering robustness:

**1. Realistic scenario:**

Phenotypic datasets (n = 180) were synthetically generated using a multivariate transformation pipeline:

- **Correlated base generation:** Traits simulated with MASS::mvrnorm() to match empirical covariance structure.
- **Distributional alignment:** Gamma-distributed traits (e.g., GRY) transformed from normal deviates via quantile matching.
- **Population structure:** Group-level covariance matrices were iteratively tuned to achieve a target phenotypic differentiation of Pst ≈ 0.15 (mean realized: 0.16 ± 0.001).
- **Heteroscedasticity:** Trait-wise standard deviations were scaled per group using empirical multipliers (e.g., Pop1 GRY SD × 0.8).

**2. Idealized control:**

A benchmark dataset featuring:

- Balanced group sizes (n = 50 per group),
- High between-group differentiation (Pst = 0.80),
- Independent traits (identity matrix),
- Homoscedastic variances.

## Clustering algorithms

We benchmarked eleven clustering methods representing major analytical paradigms:

- **Partitioning:** K-means, PAM;
- **Hierarchical:** Ward's method;
- **Model-based:** Gaussian Mixture Models (GMM);
- **Density-based:** DBSCAN, HDBSCAN;
- **Graph-theoretic:** Spectral clustering;
- **Affinity-based:** Affinity Propagation;
- **Fuzzy clustering:** Fuzzy C-means;
- **Neural-network-based:** Self-Organizing Maps (SOM);
- **Hybrid scalable:** Two-Step clustering.

For parametric methods, the number of clusters ($k = 3$) was fixed based on known population structure [1]. Non-parametric methods relied on data-driven heuristics. Parameter tuning followed best-practice guidelines (e.g., DBSCAN's eps set to 90th percentile of kNN distances; minPts = 5).

## Algorithmic settings and reproducibility

All clustering methods were applied to z-score standardized trait data using identical preprocessing and replication schemes. For algorithms requiring predefined cluster numbers, $k = 3$ was fixed based on the known population structure of fonio landraces [1]. Random seeds were controlled throughout the simulation pipeline to ensure full reproducibility.

Key parameter choices included: K-means (*nstart* = 25), DBSCAN/HDBSCAN (*minPts* = 5 with a data-driven *eps* heuristic), Fuzzy C-means (*m* = 2), Two-Step clustering (α = 0.05 trimming), and SOM (*rlen* = 100 learning steps).

No dimensionality reduction (e.g., PCA) was applied prior to clustering, as this would artificially remove the empirically calibrated inter-trait correlations that characterize real phenotypic datasets of neglected and underutilized species (NUS). The framework intentionally preserves these biological dependencies to maintain ecological realism rather than optimizing for computational convenience.

A comprehensive summary of all algorithmic settings—including package versions, R functions, and parameter specifications—is provided in S1 File (Supplementary Information). All implementations used a unified wrapper function (*apply_clustering()*), and convergence followed the default internal criteria of the respective R functions.

## Evaluation metrics

Clustering performance was evaluated using four complementary indices.

- **Adjusted Rand Index (ARI)** and **Normalized Mutual Information (NMI)** quantify external agreement between predicted and true population labels, corrected for chance [12,15].

- **Silhouette Coefficient** [26] and **Davies–Bouldin Index** [27] provide internal, label-independent assessments of cohesion and separation, enabling evaluation under weak or diffuse population structure.

Together, these external and internal metrics capture both supervised and unsupervised aspects of clustering quality, bridging algorithmic and biological interpretability.

A dispersion statistic (betadisper; [28]) was also computed to quantify within-group variability relative to population centroids.

## Statistical analysis and reproducibility

Each simulation scenario was replicated 100 times. Performance metrics were summarized using means and standard deviations; robustness was quantified as the inverse of inter-replicate variance. Pairwise comparisons employed Dunn's tests with FDR correction (α = 0.01). Visualization of clustering validity employed Principal Component Analysis (PCA) and Uniform Manifold Approximation and Projection (UMAP) [29,30].

UMAP projections were computed using the umap() function (R package *umap*, v0.2.10.0) with parameters n_neighbors = 15, min_dist = 0.1, metric = "euclidean", and a fixed random seed for reproducibility. These settings were retained because they offer a biologically sensible balance between preserving global structure and revealing local neighborhoods in medium-sized phenotypic datasets (~100–300 accessions). In UMAP, *n_neighbors* determines the balance between local versus global manifold structure; the official documentation recommends the default value of 15 as a reasonable compromise for general-purpose use, while *min_dist = 0.1* prevents excessive crowding and preserves continuous gradients typical of quantitative trait data.

These defaults are widely adopted in biological applications, and recommended practice in single-cell and phenotypic visualization places *n_neighbors* at approximately 3–10% of the sample size—corresponding closely to 15 neighbors for n ≈ 180. Accordingly, our parameterization aligns with established empirical practice and ensures stable, interpretable embeddings for phenotypic datasets of neglected crops [31–33].

All analyses were performed in R (version 4.3.0). Details of the statistical procedures, algorithms, and evaluation metrics are described above.

## Results

### Clustering performance under realistic phenotypic conditions

Under phenotypic conditions empirically calibrated from West African fonio landraces (Pst ≈ 0.16), all eleven tested clustering algorithms exhibited uniformly low performance across validation metrics (Table 1 and Fig 1). The Adjusted

**Table 1. Mean clustering metrics under realistic conditions (± SD; n = 100 unless noted).**

| Méthode | ARI | NMI | Silhouette | Davies-Bouldin |
|---|---|---|---|---|
| **GMM** | 0.053 ± 0.064 | 0.070 ± 0.062 | 0.127 ± 0.059 | 1.768 ± 0.516 |
| **TwoStep** | 0.051 ± 0.058 | 0.063 ± 0.048 | 0.188 ± 0.032 | 3.641 ± 1.347 |
| **Spectral** | 0.045 ± 0.066 | 0.055 ± 0.052 | 0.240 ± 0.057 | 1.429 ± 0.862 |
| **PAM** | 0.045 ± 0.050 | 0.058 ± 0.050 | 0.241 ± 0.025 | 1.289 ± 0.090 |
| **Fuzzy C-means** | 0.042 ± 0.047 | 0.055 ± 0.048 | 0.244 ± 0.023 | 1.288 ± 0.083 |
| **SOM** | 0.042 ± 0.045 | 0.057 ± 0.048 | 0.262 ± 0.022 | 1.223 ± 0.079 |
| **K-means** | 0.041 ± 0.044 | 0.056 ± 0.047 | 0.263 ± 0.021 | 1.223 ± 0.077 |
| **Ward.D2** | 0.040 ± 0.044 | 0.056 ± 0.045 | 0.243 ± 0.029 | 1.249 ± 0.124 |
| **Affinity Propagation** | **0.031 ± 0.019** | **0.083 ± 0.028** | 0.132 ± 0.011 | 1.556 ± 0.087 |
| **DBSCAN** | 0.007 ± 0.011 | 0.009 ± 0.008 | **0.379 ± 0.073** | 3.184 ± 2.226 |
| **HDBSCAN (n = 89)** | −0.006 ± 0.031 | 0.033 ± 0.026 | −0.153 ± 0.132 | **4.648 ± 2.169** |

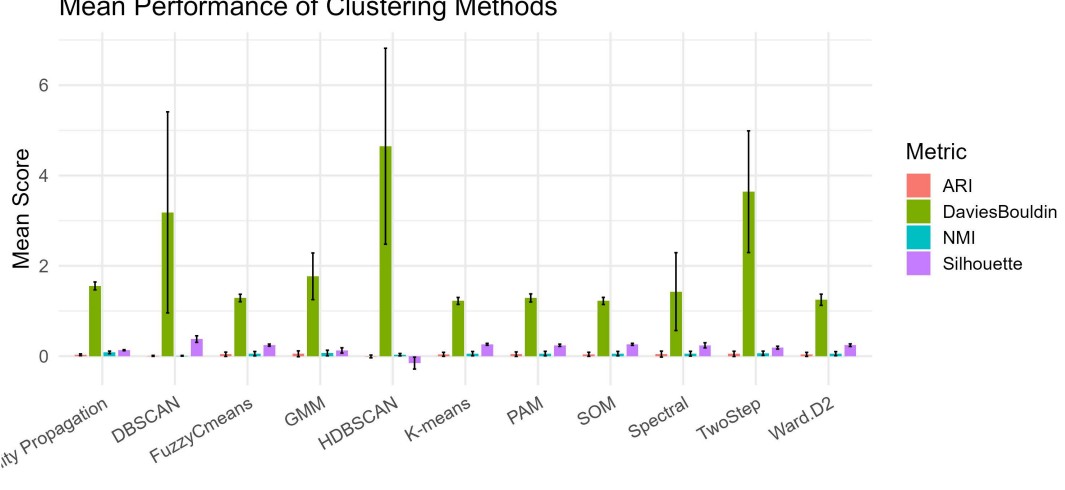

**Fig 1. Mean performance (± SD) of eleven clustering methods across ARI, NMI, Silhouette, and Davies–Bouldin indices in simulations calibrated with empirical fonio trait data (Pst ≈ 0.16).**

Rand Index (ARI) ranged from –0.006 (HDBSCAN) to 0.053 (GMM), indicating near-random agreement with true groupings. Negative ARI values indicate that agreement between predicted and true partitions is slightly lower than would be expected by chance. This occurs when clusters are statistically misaligned under weak phenotypic differentiation, and should not be interpreted as algorithmic failure but rather a signal of diffuse or overlapping biological structure. Normalized Mutual Information (NMI) values were similarly low (maximum: 0.083, Affinity Propagation), suggesting poor information retention. Silhouette coefficients exceeded the interpretability threshold (0.25) in only one case (DBSCAN: 0.379), while Davies–Bouldin indices consistently exceeded 1.2, signifying poor inter-cluster separation.

Importantly, these results were consistent across 100 simulation replicates per algorithm (except HDBSCAN: n = 89 due to convergence issues), reinforcing the reliability of observed patterns. Despite modest differences between methods, no algorithm achieved a biologically meaningful clustering outcome under realistic trait complexity and moderate population differentiation.

## Performance under idealized conditions

In the idealized scenario featuring high differentiation (Pst ≈ 0.80), independent traits, equal sample sizes, and homosce-dasticity, algorithm performance dramatically improved (Table 2 and Fig 2). Parametric methods such as GMM, Fuzzy C-means, and K-means achieved near-perfect ARI (>0.97) and NMI (>0.96) values, confirming both the integrity of the simulation pipeline and the sensitivity of these algorithms to favorable statistical conditions. Silhouette coefficients (~0.39) and Davies–Bouldin indices (~1.05) indicated well-separated, coherent clusters.

By contrast, non-parametric methods (e.g., DBSCAN, HDBSCAN, Affinity Propagation) showed lower mean perfor-mance and higher variance, suggesting reduced reliability even under ideal conditions. DBSCAN failed to converge in 4% of replicates due to poor density parameter fits.

**Table 2. Mean clustering metrics under ideal conditions (± SD; n = 100).**

| Méthode | ARI | NMI | Silhouette | Davies-Bouldin |
|---|---|---|---|---|
| GMM | 0.986 ± 0.026 | 0.981 ± 0.033 | 0.394 ± 0.077 | 1.051 ± 0.219 |
| Fuzzy C-means | 0.976 ± 0.042 | 0.968 ± 0.052 | 0.394 ± 0.076 | 1.048 ± 0.215 |
| K-means | 0.975 ± 0.045 | 0.968 ± 0.053 | 0.394 ± 0.076 | 1.047 ± 0.213 |
| Ward.D2 | 0.955 ± 0.076 | 0.949 ± 0.077 | 0.390 ± 0.080 | 1.061 ± 0.234 |
| PAM | 0.948 ± 0.073 | 0.941 ± 0.077 | 0.390 ± 0.080 | 1.058 ± 0.227 |
| Spectral | 0.937 ± 0.134 | 0.936 ± 0.119 | 0.372 ± 0.101 | 1.102 ± 0.293 |
| SOM | 0.875 ± 0.212 | 0.894 ± 0.164 | 0.371 ± 0.086 | 1.142 ± 0.292 |
| TwoStep | 0.862 ± 0.119 | 0.790 ± 0.093 | 0.340 ± 0.079 | 2.339 ± 0.478 |
| HDBSCAN | 0.652 ± 0.276 | 0.663 ± 0.210 | 0.292 ± 0.147 | 2.317 ± 1.223 |
| DBSCAN | 0.436 ± 0.391 | 0.437 ± 0.378 | 0.282 ± 0.122 | 1.775 ± 1.011 |
| Affinity Propagation | 0.415 ± 0.101 | 0.496 ± 0.069 | 0.136 ± 0.027 | 1.856 ± 0.174 |

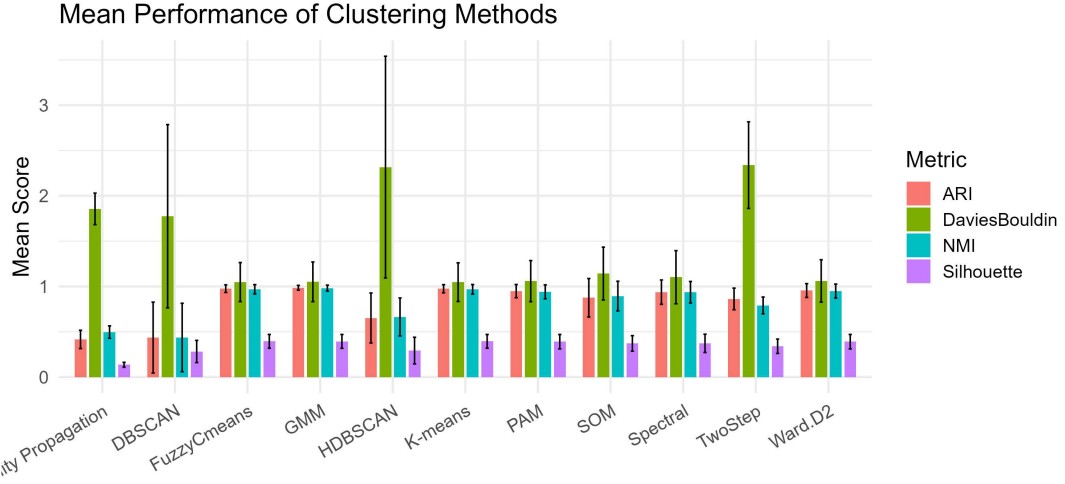

**Fig 2. Validation of simulation pipeline and algorithmic capacity under favorable statistical assumptions: independent traits, strong structure, and homoscedasticity.**

## Visual diagnostics reveal latent structure

While numerical metrics consistently suggested low clustering validity under realistic scenarios, dimension reduction techniques (PCA and UMAP) revealed subtle groupings in certain algorithms (Fig 3 and S1 File). Notably, DBSCAN and Affinity Propagation identified structure congruent with known groupings, despite low ARI/NMI scores. PCA explained >80% of variance on the first two axes, underscoring biologically relevant structure that escapes conventional metrics. This discordance highlights the limitations of relying solely on statistical indices when interpreting weakly differentiated datasets.

## Discussion

Our study presents a rigorous and biologically informed evaluation of clustering methods applied to phenotypic datasets from neglected and underutilized species (NUS), exemplified by fonio. It reveals several critical insights:

1. **Fundamental methodological mismatch:** Under realistic phenotypic simulations (Pst ≈ 0.16; group sizes 91, 43, 46), all tested clustering algorithms—ranging from partition-based (K-means, PAM) to model-based (GMM) and density-based methods (DBSCAN, HDBSCAN)—yielded uniformly poor agreement with true group labels (ARI < 0.06;

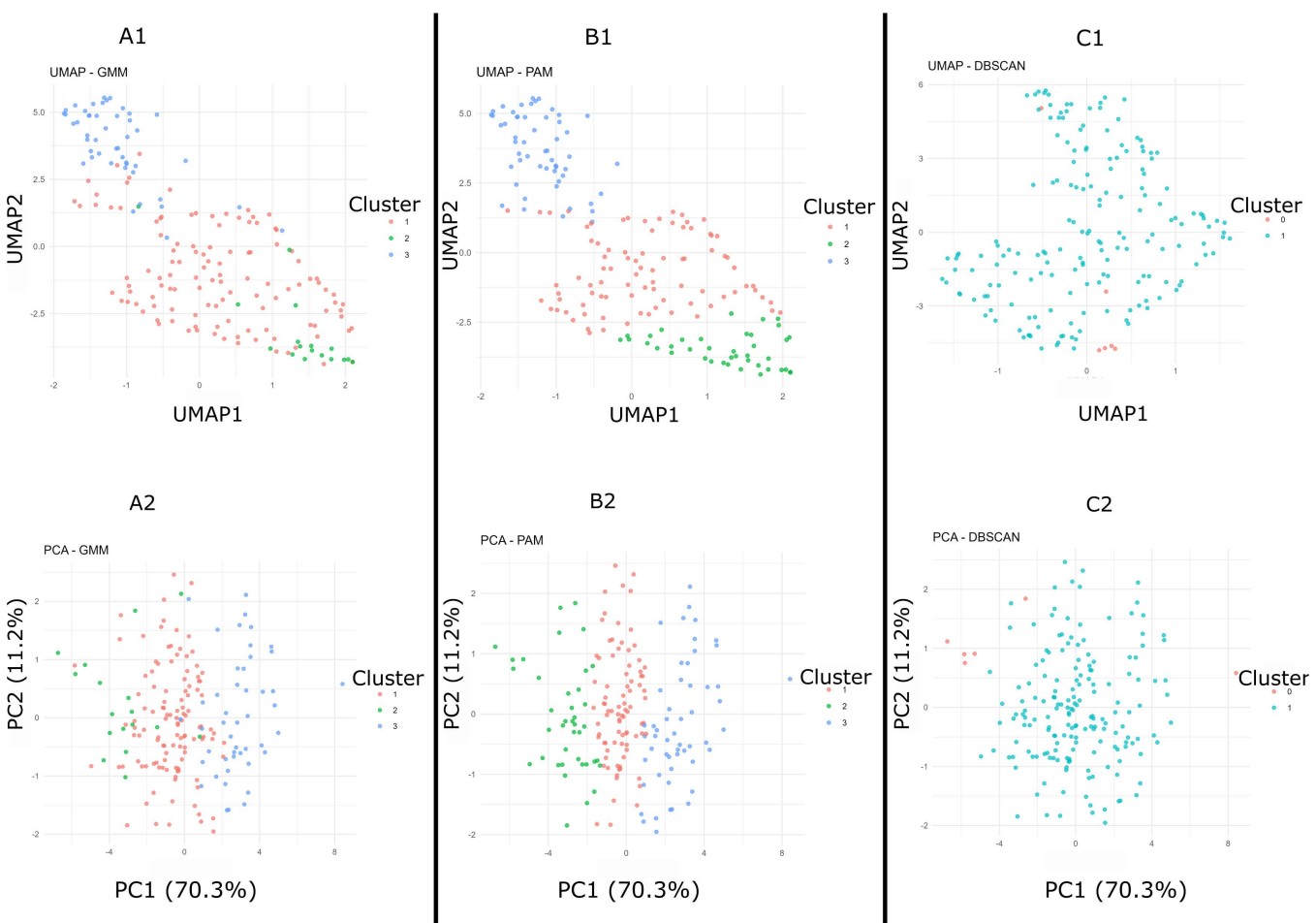

**Fig 3. PCA and UMAP reveal latent structure beyond numerical metrics.** *Visual representation of clustering patterns across dimensionality reduction methods, indicating underlying structure undetected by ARI and NMI.*

NMI < 0.09). This quantitatively demonstrates the fundamental limitations noted by Jombart et al. [6] and Odong et al. [7], extending their qualitative insights to simulation-based rigor. The small negative ARI values observed for some algorithms further confirm that, under these weakly structured conditions, agreement with the true partition can fall below what would be expected by chance alone. The reliance on assumptions of discrete, isotropic, homogeneous clusters in these algorithms is misaligned with the overlapping, correlated, heteroscedastic, and weakly differentiated structures typical of NUS phenotypic data [1,14].

2. **ARI/NMI fail to detect subtle biological structure:** Despite consistently low ARI and NMI scores, PCA and UMAP visualizations revealed coherent group separation in some clustering outputs—capturing over 80% of variance on the first two axes. This uncoupling reinforces critiques by Romano et al. [13] and Vinh et al. [12], showing that conventional metrics inherently underestimate biologically meaningful differentiation when clusters are continuous or overlapping. We argue these low scores should be viewed as signals of underlying biological complexity, rather than algorithmic failure.

3. **Visualization alone is insufficient: Diagnostics must combine metrics and plots:** Our findings underscore the necessity of integrating qualitative (PCA/UMAP) and quantitative diagnostics. Despite pronounced visual structure in DBSCAN and Affinity Propagation outputs, reliance solely on ARI/NMI would reject these methods. This aligns with critiques in single-cell genomics and suggests biodiversity analyses require multidimensional validation protocols [30,29].

4. **Impact of PCA preprocessing on clustering performance:** A common practice in applied phenotypic analyses is to perform clustering on principal component scores rather than raw traits. In our simulations, this would artificially remove the inter-trait correlations that were deliberately modeled from empirical fonio data to represent real biological dependencies. While PCA-based preprocessing may improve numerical stability or visual interpretability, it would not resolve the fundamental issue of weak phenotypic differentiation (Pst ≈ 0.15). Therefore, we expect that clustering on PCA-reduced data would yield marginally higher scores but would not alter the overall conclusion that conventional algorithms fail under biologically realistic conditions.

5. **Trade-off between stability and accuracy:** Analyzing robustness (inverse ARI variance) exposes a critical practical trade-off: high-performing but unstable methods (e.g., GMM) versus more stable, interpretable algorithms (e.g., PAM, Ward's hierarchical). This finding supports prioritizing reproducibility and interpretability in applied NUS research—a stance aligned with Pilling et al. [5] and Vodouhè et al. [4].

6. **Toward data-driven clustering via empirical simulation:** Crucially, our framework demonstrates the value of context-specific simulation. By parameterizing distributions (normal, gamma, binomial), sample imbalance, correlation structures, heteroscedasticity, and group number, researchers can identify clustering methods most robust to specific dataset properties and assumption violations. This moves phenotypic clustering from a one-size-fits-all approach to a tailored, data-driven strategy.

7. **Need for novel clustering validation metrics:** The observed mismatch between conventional validation metrics and visual/biological structure highlights the urgent necessity for new indices—designed for weakly structured NUS data. Such metrics must integrate geometric separation, internal cohesion, and be robust to non-normality, unbalanced designs, heteroscedasticity, and nonlinear correlations.

8. **Mapping performance across population differentiation dradient:** Future work should focus on how clustering detectability varies with phenotypic differentiation. As suggested by Jombart et al. [6], critical $P_{st}$ thresholds may exist below which conventional clustering algorithms become unreliable. The present study concentrates on a single empirically grounded differentiation level ($P_{st} \approx 0.15$), but future analyses should systematically map clustering performance across a continuum of differentiation values.

Empirically identifying these detectability boundaries would help practitioners to:

1. Distinguish biologically diffuse population structure from methodological artefacts;

2. Establish data-driven guidelines for the applicability and interpretation of clustering analyses in NUS research; and

3. Develop generalizable frameworks that explicitly account for differentiation level rather than relying on one-size-fits-all approaches.

Such gradient-based analyses represent a natural extension of the current simulation framework, which can be readily parameterized to explore $P_{st}$ values ranging from near-neutral differentiation ($P_{st} \approx 0.05$) to strong divergence ($P_{st} > 0.25$). This line of research will form the basis of future work aimed at quantifying the relationship between clustering detectability and the intensity of phenotypic differentiation.

9. **Broad impact and future extensions:** Our study challenges current NUS phenotypic research paradigms—routinely reliant on K-means, GMM, or DBSCAN. Instead, we propose a robust methodological reform grounded in empirically calibrated simulations, combined validation diagnostics, and context-aware algorithm selection. This paradigm is generalizable to NUS with varying sample sizes, trait distributions, complex correlations, and population structures. Future extensions should also incorporate explicit handling of missing data mechanisms (MCAR/MAR), extreme sample imbalance, continuous differentiation gradients, and integration with genomic validation [34], following the statistical principles outlined by Little & Rubin [35].

10. **Toward biologically meaningful clustering metrics:** Our results confirm that standard external indices such as ARI and NMI rapidly lose discriminative power when phenotypic differentiation is weak (Pst<0.2). This limitation, already noted in population genomics and community ecology [15,12], highlights the need for alternative, biologically informed criteria.

The inclusion of internal indices such as Silhouette and Davies–Bouldin partially alleviates this issue by quantifying within-group cohesion and between-group separation without reference to true labels. However, these metrics remain geometry-based and do not account for biological plausibility or trait redundancy.

Future developments should therefore explore hybrid approaches — combining statistical validation with biological constraints (e.g., trait heritability, ecological coherence, or phenotypic dispersion; [28]) — to define more meaningful clustering diagnostics in agrobiodiversity research.

11. **Practical implications for NUS research and conservation:** Practical recommendations for biodiversity practitioners and breeders before clustering:

   • **Calibrate expectations via simulation.** Use the R framework provided (Zenodo DOI: [10.5281/zenodo.15877863]) to benchmark expected algorithmic performance under your crop's specific trait architecture before analyzing real data. Parameterizing simulations with empirical trait means, variances, correlations, and heteroscedasticity enables realistic assessment of clustering detectability.

*During analysis:*

• **Combine multiple validation approaches.** Supplement ARI and NMI with internal cohesion metrics (Silhouette, Davies–Bouldin), visual diagnostics (PCA, UMAP), and biological expertise. Integrated interpretation mitigates over-reliance on purely statistical indices, especially in weakly differentiated datasets.

*For decision-making:*

• **Prioritize reproducibility over peak performance.** When clusters inform breeding or conservation actions, favor stable and interpretable algorithms (e.g., PAM, Ward) that yield consistent results across resampling rather than complex models that are numerically optimal but unstable (e.g., GMM).

*For methodological development:*

- **Establish detectability thresholds.** Use the simulation platform to explore how clustering performance changes with increasing phenotypic differentiation (Pst), identifying thresholds where structure becomes reliably detectable. Such detectability curves can guide algorithm choice and interpretation in other neglected crops.

We encourage practitioners to apply this diagnostic framework before making conservation or breeding decisions based on phenotypic clustering—particularly for NUS where genomic validation may be unavailable. This approach bridges methodological rigor and applied relevance, strengthening the reliability of phenotypic inferences in biodiversity research.

## Conclusion

This study demonstrates that conventional clustering algorithms systematically underperform when applied to realistically simulated phenotypic data from neglected and underutilized species (NUS), such as fonio. These results reveal not only methodological limitations, but also a deeper misalignment between algorithmic assumptions and the biological realities of NUS datasets—characterized by overlapping traits, heteroscedasticity, and modest differentiation.

Critically, our analyses show that low ARI and NMI scores can obscure meaningful biological structure, which remains detectable through visual and geometric diagnostics such as PCA and UMAP. This highlights the need to treat clustering as an exploratory rather than confirmatory tool, combining quantitative indices with qualitative and biological interpretation.

The biologically calibrated simulation framework developed here offers a reproducible, flexible diagnostic platform to benchmark clustering performance under realistic conditions. It enables researchers to align methodological choices with empirical data properties, fostering more robust and interpretable phenotypic stratification.

Future research should focus on developing biologically grounded validation metrics better suited to continuous and weakly structured biological data, and on systematically mapping clustering performance across gradients of phenotypic differentiation (Pst). Integrating missing-data mechanisms (MCAR/MAR) and genomic validation will further strengthen the framework's applicability to real-world datasets.

Ultimately, this simulation-based diagnostic platform is crop-agnostic and extensible across species, providing a foundation for data-driven standards in phenotypic structure detection. By bridging methodological rigor with biological realism, it offers a path toward more transparent, reproducible, and evolutionarily meaningful biodiversity assessment in data-scarce crops.

## Supporting information

**S1 File. Supplementary figures and table.** This compressed archive contains Figures S1–S12 and Table S1.
(ZIP)

## Acknowledgments

The author expresses sincere appreciation to Dr. Devra I. Jarvis and the Raffaella Foundation for their continuous efforts to promote research and capacity building on agrobiodiversity in West Africa and beyond. The present work aligns with this broader vision, although it was conducted independently within the author's academic affiliation at the University Abdou Moumouni.

## Author contributions

**Conceptualization:** Abdel Kader Naino Jika.

**Data curation:** Abdel Kader Naino Jika.

**Formal analysis:** Abdel Kader Naino Jika.

**Investigation:** Abdel Kader Naino Jika.

**Methodology:** Abdel Kader Naino Jika.

**Project administration:** Abdel Kader Naino Jika.

**Software:** Abdel Kader Naino Jika.

**Validation:** Abdel Kader Naino Jika.

**Visualization:** Abdel Kader Naino Jika.

**Writing – original draft:** Abdel Kader Naino Jika.

**Writing – review & editing:** Abdel Kader Naino Jika.

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
