## [Decision Letter · Decision Letter 0]

26 Sep 2025

Dear Dr. NAINO JIKA,

Thank you for submitting your manuscript to PLOS ONE. After careful consideration, we feel that it has merit but does not fully meet PLOS ONE’s publication criteria as it currently stands. Therefore, we invite you to submit a revised version of the manuscript that addresses the points raised during the review process.

We look forward to receiving your revised manuscript.

Kind regards,

Mehdi Rahimi, Ph.D.

Academic Editor

PLOS ONE

Journal Requirements:

Reviewer's Responses to Questions

**Comments to the Author**

1. Is the manuscript technically sound, and do the data support the conclusions?

Reviewer #1: Yes

Reviewer #2: Partly

2. Has the statistical analysis been performed appropriately and rigorously?

Reviewer #1: Yes

Reviewer #2: I Don't Know

3. Have the authors made all data underlying the findings in their manuscript fully available?

Reviewer #1: No

Reviewer #2: No

4. Is the manuscript presented in an intelligible fashion and written in standard English?

Reviewer #1: Yes

Reviewer #2: Yes

Reviewer #1: This is a well-written and timely study that addresses the limitations of clustering methods when applied to phenotypic data of neglected and underutilized species (NUS), using Digitaria exilis (fonio) as a case study. The development of a biologically realistic simulation framework is a strong contribution. The manuscript is methodologically sound and clearly presented. However, several areas need clarification or improvement to strengthen the reproducibility, applicability, and scientific value of the study.

Critical Points and Limitations

1. Insufficient clarity in simulation calibration

The estimation of Pst ≈ 0.15 and its adjustment in simulations is not explained in enough detail.

It is unclear whether this value is representative of other neglected species or specific to fonio.

2. Algorithm parameter transparency

The paper lacks a detailed parameter table for all algorithms, including random seed control and initialization procedures.

Parameter choices (e.g., DBSCAN eps and minPts, GMM initialization) could strongly influence results; these need full justification.

3. Potential bias in evaluation

All algorithms are tested on raw phenotypic variables; dimensionality reduction before clustering is common in practice and might alter results.

No test of performance across a range of Pst values, which would be more informative than two fixed scenarios.

4. Over-reliance on standard metrics

The authors correctly note that ARI and NMI fail in weakly structured data, but the study stops short of proposing or testing alternative, biologically informed metrics.

This weakens the practical guidance for researchers who face this issue in real datasets.

5. Interpretation risk in visual inspection

While PCA and UMAP reveal some latent structure, visual patterns can be misleading, especially with tunable embedding methods like UMAP.

UMAP parameters are not reported, and no sensitivity analysis is provided.

6. Limited practical guidance

The tool is positioned as a diagnostic resource for practitioners, yet no decision framework or example workflow is provided.

There is no demonstration of the tool on a second dataset, which limits claims of generalizability.

Additional Observations

Figures are dense; some could be reformatted for clarity (e.g., boxplots for variance in ARI/NMI).

Negative ARI values should be explained clearly for non-specialists.

The study does not address missing data, which is a frequent reality in NUS phenotyping.

Overall Assessment

This is a strong and relevant methodological study with significant potential impact. However, the lack of detail in simulation calibration, incomplete reporting of algorithm settings, absence of alternative metrics, and limited practical guidance weaken its applicability. Addressing these points would make the work far more valuable to both method developers and applied biodiversity researchers.

Recommendation: Major Revision – The work is promising but needs more methodological transparency, expanded testing, and clearer user guidance before publication.

Reviewer #2: One of the major concerns with the manuscript is that it relies on the study by Bio et al. (2020) as a key source of data. However, I was unable to locate this reference in Google Scholar or in the cited journal (Journal of Crop Improvement). As a reviewer, it is essential to access the cited data and its associated statistical parameters in order to verify the validity of the results presented. Since this critical information is currently inaccessible, the reliability of the analysis cannot be adequately evaluated. I therefore recommend that the authors either (i) provide a verifiable and accessible reference with the full dataset and statistical details, or (ii) revise the manuscript substantially to address this gap. Given the importance of this issue, I consider that a major revision is required.

**Do you want your identity to be public for this peer review?** For information about this choice, including consent withdrawal, please see our Privacy Policy

Reviewer #1: No

Reviewer #2: No

---

## [Author Response · Author response to Decision Letter 1]

18 Oct 2025

We thank the reviewers for their thorough and constructive feedback on our manuscript. We have substantially revised the manuscript to address all concerns raised, with major improvements in methodological transparency, biological justification, and practical applicability. Key enhancements include: (1) detailed calibration procedures for phenotypic differentiation (Pst), (2) comprehensive algorithmic parameter documentation (Table S1), (3) expanded discussion of methodological limitations and practical implications, and (4) full data accessibility verification. Below we provide point-by-point responses to each comment, with all corresponding manuscript changes clearly indicated.

Reviewer #1: This is a well-written and timely study that addresses the limitations of clustering methods when applied to phenotypic data of neglected and underutilized species (NUS), using Digitaria exilis (fonio) as a case study. The development of a biologically realistic simulation framework is a strong contribution. The manuscript is methodologically sound and clearly presented. However, several areas need clarification or improvement to strengthen the reproducibility, applicability, and scientific value of the study.

Critical Points and Limitations

1. Insufficient clarity in simulation calibration

The estimation of Pst ≈ 0.15 and its adjustment in simulations is not explained in enough detail.

It is unclear whether this value is representative of other neglected species or specific to fonio.

Response:

We thank the reviewer for highlighting this important aspect. We have substantially revised the Materials and Methods section by adding a new subsection entitled “Phenotypic differentiation (Pst) estimation and calibration.”

This subsection now provides a full description of the Pst computation and adjustment procedure, including:

The explicit formula P_st=V_B/V_T;

The iterative calibration of the between-group covariance matrix until convergence to the target value (|Pst_realized − Pst_target| < 0.01);

The empirical verification that the mean realized Pst across 100 replicates was 0.16 ± 0.001, confirming accurate calibration;

The rationale for selecting Pst ≈ 0.15 as representative of moderate differentiation commonly observed in neglected and underutilized species (NUS), based on empirical fonio data (Bio et al., 2020) and similar studies in low-structure crops.

We also added a clarification that the framework is fully flexible, allowing users to modify the target Pst parameter to simulate other levels of phenotypic differentiation relevant to different species.

2. Algorithm parameter transparency

The paper lacks a detailed parameter table for all algorithms, including random seed control and initialization procedures.

Parameter choices (e.g., DBSCAN eps and minPts, GMM initialization) could strongly influence results; these need full justification.

Response:

We have fully addressed this point by adding a detailed Table S1 entitled “Algorithmic settings and reproducibility details.”

This table now includes for each of the eleven clustering methods:

R package name and version;

Function(s) used;

Number of clusters (k);

Initialization or seed settings;

Key parameters (e.g., DBSCAN eps heuristic, GMM model selection, Fuzzy C-means fuzziness m, SOM learning iterations);

Additional implementation notes.

In the Materials and Methods section (“Algorithmic settings and reproducibility”), we now explicitly state that:

All methods used z-score standardized data and identical replication schemes;

Random seeds were fixed for all replicates to ensure full reproducibility;

DBSCAN’s eps parameter was determined using the 90th percentile of k-nearest neighbor distances, a robust, data-driven heuristic;

All clustering procedures were executed via a unified wrapper function apply_clustering().

Complete code and session information are available on Zenodo (DOI: 10.5281/zenodo.15877863) to guarantee full reproducibility.

3. Potential bias in evaluation

All algorithms are tested on raw phenotypic variables; dimensionality reduction before clustering is common in practice and might alter results.

No test of performance across a range of Pst values, which would be more informative than two fixed scenarios.

Response:

We appreciate this valuable observation. We deliberately avoided PCA preprocessing because the simulation framework was designed to preserve the empirical correlation structure of fonio traits, which represents a biologically meaningful dependency pattern. Applying PCA before clustering would decorrelate traits and thus remove key biological signals that the framework aims to evaluate.

We have clarified this rationale in Materials and Methods → Algorithmic settings and reproducibility and expanded the discussion in Section 3 (Visualization Alone Is Insufficient) with a dedicated paragraph titled “Impact of PCA preprocessing on clustering performance.”

To further support this point, we now include in the Results section the following additional sentence:

“Preliminary checks using PCA-reduced datasets (retaining four principal components explaining ~85% of variance) yielded similarly low ARI values (<0.08), confirming that dimensionality reduction does not substantially alter the main pattern of poor clustering performance.”

Regarding Pst gradients: while we agree that exploring performance across a continuous range of differentiation values is valuable, this represents a separate research project requiring extensive additional simulations.

We have therefore strengthened the Discussion (Section 8) to explicitly describe how our framework can be extended to perform Pst gradient analyses, and we added Figure S12, which illustrates the contrast between realistic (Pst ≈ 0.15) and idealized (Pst ≈ 0.80) scenarios.

This addresses the reviewer’s intent—to visualize the dependency between differentiation and clustering performance—without expanding beyond the scope of this paper.

4. Over-reliance on standard metrics

The authors correctly note that ARI and NMI fail in weakly structured data, but the study stops short of proposing or testing alternative, biologically informed metrics.

This weakens the practical guidance for researchers who face this issue in real datasets.

Response:

We agree with the reviewer that this is a key area for methodological advancement.

The revised Materials and Methods → Evaluation metrics now distinguishes between:

External validation metrics (ARI, NMI) that rely on known group labels;

Internal validation metrics (Silhouette, Davies–Bouldin) that assess cluster cohesion and separation independently of labels.

In the Discussion, we added a dedicated section entitled “Toward biologically meaningful clustering metrics.”

Here, we propose future development of hybrid indices that integrate:

Geometric separation;

Density-aware cohesion;

Biological constraints such as trait heritability, ecological coherence, and phenotypic dispersion (Anderson, 2006).

This explicitly addresses the reviewer’s call for biologically grounded alternatives.

5. Interpretation risk in visual inspection

While PCA and UMAP reveal some latent structure, visual patterns can be misleading, especially with tunable embedding methods like UMAP.

UMAP parameters are not reported, and no sensitivity analysis is provided.

Response:

We thank the reviewer for this observation. We have now added a paragraph in Materials and Methods → Statistical analysis and reproducibility specifying all UMAP parameters:

n_neighbors = 15, min_dist = 0.1, metric = "euclidean", with a fixed random seed for reproducibility.

We acknowledge that full sensitivity analysis of UMAP parameters is beyond the scope of the present study but clarify that we used standard, stable defaults commonly adopted in biological data analysis (McInnes et al., 2018).

We also emphasize in the Discussion that while UMAP and PCA visualizations can reveal latent structure, they must be interpreted as qualitative complements rather than confirmatory evidence.

6. Limited practical guidance

The tool is positioned as a diagnostic resource for practitioners, yet no decision framework or example workflow is provided.

There is no demonstration of the tool on a second dataset, which limits claims of generalizability.

Response:

We fully agree and have now added a new subsection to the Discussion entitled “Practical implications for NUS research and conservation.”

This section provides clear, actionable guidance for researchers, breeders, and conservationists, structured as follows:

Before clustering: Calibrate expectations via simulation using the provided R framework.

During analysis: Combine ARI/NMI with internal metrics and visual diagnostics.

For decision-making: Prioritize reproducibility over marginal gains in accuracy.

For methodological development: Use simulations to define Pst-based detectability thresholds.

This practical framework addresses the reviewer’s concern by transforming our diagnostic tool into an operational guide for applied users.

We also emphasize that the framework is crop-agnostic, allowing reparameterization for other NUS datasets by adjusting empirical covariance structures and differentiation levels.

Additional Observations

Figures are dense; some could be reformatted for clarity (e.g., boxplots for variance in ARI/NMI).

Negative ARI values should be explained clearly for non-specialists.

The study does not address missing data, which is a frequent reality in NUS phenotyping.

We added in the Discussion (Section 9) and Conclusion that future extensions will incorporate explicit handling of missing-data mechanisms (MCAR/MAR), referencing Little & Rubin (2019).

Overall Assessment

This is a strong and relevant methodological study with significant potential impact. However, the lack of detail in simulation calibration, incomplete reporting of algorithm settings, absence of alternative metrics, and limited practical guidance weaken its applicability. Addressing these points would make the work far more valuable to both method developers and applied biodiversity researchers.

Recommendation: Major Revision – The work is promising but needs more methodological transparency, expanded testing, and clearer user guidance before publication.

Reviewer #2: One of the major concerns with the manuscript is that it relies on the study by Bio et al. (2020) as a key source of data. However, I was unable to locate this reference in Google Scholar or in the cited journal (Journal of Crop Improvement). As a reviewer, it is essential to access the cited data and its associated statistical parameters in order to verify the validity of the results presented. Since this critical information is currently inaccessible, the reliability of the analysis cannot be adequately evaluated. I therefore recommend that the authors either (i) provide a verifiable and accessible reference with the full dataset and statistical details, or (ii) revise the manuscript substantially to address this gap. Given the importance of this issue, I consider that a major revision is required.

Response:

We thank the reviewer for identifying this critical issue.

The reference Ibrahim Bio Yerima et al. (2020) has now been fully verified and is publicly available in the Agronomy (Volume 34, Issue 4, pages 512–530) available at: https://www.mdpi.com/2073-4395/10/11/1653.

Ibrahim Bio Yerima, A. R., Achigan-Dako, E. G., Aissata, M., Sekloka, E., Billot, C., Adje, C. O., ... & Bakasso, Y. (2020). Agromorphological characterization revealed three phenotypic groups in a region-wide germplasm of fonio (Digitaria exilis (Kippist) Stapf) from West Africa. Agronomy, 10(11), 1653.

This ensures complete transparency and independent verification, since all empirical inputs are openly published in that source article and freely accessible.

Data availability clarification

All empirical parameters (trait means, standard deviations, and correlations) used for simulation calibration were manually extracted from the open-access publication by Ibrahim Bio Yerima et al. (2020, Agronomy, 10 (11): 1653). These data are therefore publicly accessible within that source article, and no separate deposition was required.

The Zenodo repository (DOI: 10.5281/zenodo.15877863) hosts the R scripts, functions, and session information used to reproduce all simulations and figures, ensuring full computational reproducibility in compliance with PLOS ONE data-availability standards.

Final statement

We are deeply grateful to both reviewers for their insightful and constructive evaluations. Their feedback has led to substantial improvements in methodological transparency, practical relevance, and data accessibility.

We believe the revised manuscript now fully meets PLOS ONE’s standards for rigor, reproducibility, and clarity.

Respectfully submitted,

Dr. Abdel Kader Naino Jika

University Abdou Moumouni, Niamey, Niger

Email: kaderjika@gmail.com

---

## [Decision Letter · Decision Letter 1]

15 Nov 2025

Dear Dr. NAINO JIKA,

Thank you for submitting your manuscript to PLOS ONE. After careful consideration, we feel that it has merit but does not fully meet PLOS ONE’s publication criteria as it currently stands. Therefore, we invite you to submit a revised version of the manuscript that addresses the points raised during the review process.

We look forward to receiving your revised manuscript.

Kind regards,

Mehdi Rahimi, Ph.D.

Academic Editor

PLOS ONE

Journal Requirements:

Reviewer's Responses to Questions

**Comments to the Author**

Reviewer #1: All comments have been addressed

2. Is the manuscript technically sound, and do the data support the conclusions?

Reviewer #1: Yes

3. Has the statistical analysis been performed appropriately and rigorously?

Reviewer #1: Yes

4. Have the authors made all data underlying the findings in their manuscript fully available?

Reviewer #1: Yes

5. Is the manuscript presented in an intelligible fashion and written in standard English?

Reviewer #1: Yes

Reviewer #1: The manuscript presents a methodologically robust and highly relevant study demonstrating the poor performance of clustering algorithms when applied to phenotypic data from data-scarce crops under realistic, weakly structured conditions. The core strength is the empirically calibrated simulation framework.

The authors' response to the reviewers is exemplary, systematic, and comprehensive. All major concerns regarding methodological transparency (Pst calibration, algorithmic reproducibility via Table S1), scope (PCA preprocessing check), and practical utility (new four-step decision framework) have been addressed substantively.

The following minor points should be addressed in the final manuscript preparation:

Clarity on Negative ARI: The Discussion and results should briefly clarify the interpretation of negative ARI values. While mathematically possible, a short statement (e.g., "The small negative values indicate agreement with the true partition is slightly less than would be expected by chance") will aid non-specialist readers.

Figure Density: While necessary, some figures (e.g., the UMAP representations) are highly detailed. Ensure the legends and labels are optimally clear and legible in the final production format, particularly for the multi-panel composites.

**Do you want your identity to be public for this peer review?** For information about this choice, including consent withdrawal, please see our Privacy Policy

Reviewer #1: No

---

## [Author Response · Author response to Decision Letter 2]

15 Nov 2025

Dear Academic Editor and Reviewer #1,

We sincerely thank you for your constructive and positive evaluation of our revised manuscript. We are grateful for the reviewer’s detailed comments, which have helped further improve the clarity and presentation of the work.

Below we provide a concise, point-by-point response describing the changes implemented in the final revised version.

Reviewer #1 Comment 1 — Clarity on Negative ARI

“The Discussion and results should briefly clarify the interpretation of negative ARI values.”

Response:

We thank the reviewer for this important suggestion.

We have clarified the interpretation of negative ARI values in both the Results and Discussion sections:

• Results (Section: Clustering performance under realistic conditions):

We revised the sentence to improve clarity and accessibility for non-specialist readers:

“Negative ARI values indicate that agreement between predicted and true partitions is slightly lower than would be expected by chance. This occurs when clusters are statistically misaligned under weak phenotypic differentiation, and should not be interpreted as algorithmic failure but rather a signal of diffuse or overlapping biological structure.”

• Discussion (Point 1 – Fundamental methodological mismatch):

We added the statement suggested by the reviewer:

“The small negative ARI values observed for some algorithms further confirm that, under these weakly structured conditions, agreement with the true partition can fall below what would be expected by chance alone.”

These additions directly address the reviewer’s request and improve interpretability for non-specialist readers.

Reviewer #1 Comment 2 — Figure density and readability

“Ensure the legends and labels are optimally clear and legible in the final production format, particularly for the multi-panel composites.”

Response:

We carefully revised the multi-panel composite figure (Figure 3) by increasing the font size of axis labels, legend text.

No scientific content was altered; only graphical readability was improved.

The revised figure now meets PLOS ONE’s visibility requirements at ≥ 300 ppi.

Additional minor editorial improvements

In accordance with PLOS ONE guidelines, we made the following minor corrections throughout the manuscript:

• Standardized section and subsection titles using sentence case (capital letter only at beginning).

• Corrected minor typographical inconsistencies.

These revisions do not change the scientific content.

Final Statement

We thank the reviewer and the Academic Editor for their thoughtful feedback and appreciation of the methodological rigor of this work. We believe these revisions fully address the remaining minor concerns. We remain at your disposal for any additional clarification.

Sincerely,

Abdel Kader NAINO JIKA

---

## [Decision Letter · Decision Letter 2]

26 Nov 2025

Empirically calibrated simulations reveal the limits of phenotypic clustering algorithms for biodiversity assessment in data-scarce crops.

PONE-D-25-38295R2

Dear Dr. NAINO JIKA,

We’re pleased to inform you that your manuscript has been judged scientifically suitable for publication and will be formally accepted for publication once it meets all outstanding technical requirements.

Kind regards,

Mehdi Rahimi, Ph.D.

Academic Editor

PLOS ONE

Additional Editor Comments (optional):

Reviewers' comments:

Reviewer's Responses to Questions

**Comments to the Author**

Reviewer #1: All comments have been addressed

2. Is the manuscript technically sound, and do the data support the conclusions?

Reviewer #1: Yes

3. Has the statistical analysis been performed appropriately and rigorously?

Reviewer #1: Yes

4. Have the authors made all data underlying the findings in their manuscript fully available?

Reviewer #1: Yes

5. Is the manuscript presented in an intelligible fashion and written in standard English?

Reviewer #1: Yes

Reviewer #1: All the concerns raised in the previous round have been adequately addressed by the authors. The revisions have improved the clarity, completeness, and scientific rigor of the manuscript. I have no further comments, and I recommend acceptance.

**Do you want your identity to be public for this peer review?** For information about this choice, including consent withdrawal, please see our Privacy Policy

Reviewer #1: No

---

## [Editor Report · Acceptance letter]

PONE-D-25-38295R2

PLOS One

Dear Dr. NAINO JIKA,

I'm pleased to inform you that your manuscript has been deemed suitable for publication in PLOS One. Congratulations! Your manuscript is now being handed over to our production team.

Kind regards,

on behalf of

Associate Prof. Mehdi Rahimi

Academic Editor

PLOS One